# Advances in Anti-Cancer Activities of Flavonoids in *Scutellariae radix*: Perspectives on Mechanism

**DOI:** 10.3390/ijms231911042

**Published:** 2022-09-20

**Authors:** Yiqing Gu, Qi Zheng, Guifang Fan, Runping Liu

**Affiliations:** School of Chinese Materia Medica, Beijing University of Chinese Medicine, 11 Bei San Huan Dong Road, Beijing 100029, China

**Keywords:** baicalein, baicalin, wogonin, wogonoside, anti-cancer activity

## Abstract

Despite encouraging progresses in the development of novel therapies, cancer remains the dominant cause of disease-related mortality and has become a leading economic and healthcare burden worldwide. *Scutellariae radix* (SR, *Huangqin* in Chinese) is a common herb used in traditional Chinese medicine, with a long history in treating a series of symptoms resulting from cancer, like dysregulated immune response and metabolic abnormalities. As major bioactive ingredients extracted from SR, flavonoids, including baicalein, wogonin, along with their glycosides (baicalin and wogonoside), represent promising pharmacological and anti-tumor activities and deserve extensive research attention. Emerging evidence has made great strides in elucidating the multi-targeting therapeutic mechanisms and key signaling pathways underlying the efficacious potential of flavonoids derived from SR in the field of cancer treatment. In this current review, we aim to summarize the pharmacological actions of flavonoids against various cancers in vivo and in vitro. Moreover, we also make a brief summarization of the endeavor in developing a drug delivery system or structural modification to enhance the bioavailability and biological activities of flavonoid monomers. Taken together, flavonoid components in SR have great potential to be developed as adjuvant or even primary therapies for the clinical management of cancers and have a promising prospect.

## 1. Introduction

Cancer remains a major public health issue worldwide and is well-recognized as a leading cause of morbidity and mortality, second only to cardiovascular disease [1]. The incidence of newly diagnosed cancer keeps raising and the overall mortality rate is concurrently increasing, estimated to rise from 6.1 million in 2008 to 10.6 million in 2030 in terms of global demographic characteristics [2]. A tumor is an abnormal growth of cells that serves no functional purposes but has the potential to invade or spread to adjacent tissues and other parts of the body, and can be observed in almost all organs, including blood, breast, lung, prostate, colon, liver, pancreas, and brain [3,4]. Aberrant cell proliferation and survival are the root cause of cancer. The process of carcinogenesis, including initiation, promotion, progression, and metastasis, is a complex interplay between multiple factors, such as genetic determinants and epigenetic alterations [5], as well as nourishment in the tumor microenvironment (TME). Over the past several decades, a variety of approaches, including aggressive surgery, chemotherapy, radiotherapy, targeted therapy, and innovative immune therapy, were developed to manage the development of cancers. However, numerous limitations and side effects of these therapies, either alone or in combination with each other, have been comprehensively reported: (1) Low response rate to certain targeted therapies due to the dramatic heterogenicity of cancers. (2) The increasing incidence of resistance and multi-resistance to chemo- and targeted therapies. (3) Severe adverse effects, such as myelosuppression, gastrointestinal tract symptoms, liver and renal toxicity, and cardiac damage. Although immunotherapies have become emerging trends in the study of novel anti-tumor strategies in recent years, inconsistent outcomes in different cancers and treatment-related lethal cytokine storms tend to significantly obscure its clinical application. Thus, the exploitation of innovative and effective cancer therapies or adjuvant treatments without compromising therapeutic efficacy, but improving both the survival and quality of patients’ life, have been and remain the major challenge of anti-cancer treatment [6].

Emerging evidence has suggested that traditional Chinses medicine (TCM) and considerable natural products derived from TCM herbs are promising candidates for cancer and associated complications. The progress in identifying natural products in TCM provides a bright new chemical library for anti-cancer drug discovery. *Scutellariae radix* (SR)*,* also known as *Chinese skullcap* or *Huangqin* in Chinese, is the dry root of *Scutellaria baicalensis* Georgi, which is a flowering perennial herb of the *Lamiaceae* family and primarily distributed in Europe, North America, East Asia, and South America such as Mongolia, Korea, and Russian and Siberia [7]. SR was firstly described in *Shen Nong’s Materia Medica*, and had been repeatedly cited by all the later TCM monographs for more than 2000 years [8]. SR is traditionally prescribed for clearing heat, removing dampness, purging fire, detoxification, anti-inflammation, and promoting digestion. Modern pharmacological research mainly focuses on elucidating its significant protective effects against inflammation, bacteria, and virus. Studies have shown that flavonoids, and to date, a total of 126 small molecule compounds and 6 polysaccharides have been isolated from SR. Ever since baicalein was isolated in 1889, more than 40 flavonoids have been separated and identified and are the most abundant components in SR [8,9]. Among them, baicalein, wogonin, along with their glycosides (baicalin and wogonoside) are well-established as the most potent drug candidates. In clinical practice, hot water is generally used for extraction. According to statistics, the yield of baicalin is usually about 26%, wogonin-glucuronide 10%, baicalin 2%, and wogonin 0.2% [6]. Emerging evidence has suggested that flavonoids, such as baicalein and wogonin, exert promising antitumor pharmacological activity both in vitro in cancer cell lines as well as in vivo in animal models, and the underlying mechanisms of action include the induction of apoptosis and tumor cell cycle arrest, inhibition of tumor migration and invasion, suppression of tumor angiogenesis, and overcoming drug resistance. In this current review, we aim to provide a comprehensive summary of recent studies regarding the potential application of bioactive flavonoids derived from SR as prospective therapeutic options for the treatment of cancers [10]. Mechanisms and molecular targets involved in the anticancer activity are highlighted to shed inspiring light on the discovery of more effective and safer derivatives.

## 2. Induction of Cancer Cell Apoptosis by Major Flavonoids in SR

Apoptosis is an ordered and programmed cellular process where cells actively pursue a course toward death triggered by certain stimulus [11]. The de-regulated apoptotic system allows malignant cells to escape from this program, leading to uncontrolled cancer cell proliferation which ultimately results in tumor growth [12]. There are two pathways by which apoptosis can be triggered, involving the activation of either intrinsic signals or extrinsic pathways [13]. The intrinsic apoptotic pathway involves a change in mitochondrial permeability regulated by pro-apoptotic, such as BCL2-Associated X, Bcl-2asociated death promoter, and anti-apoptotic B-cell lymphoma 2 (Bcl-2) family, such as Myeloid cell leukemia-1 (Mcl-1), Bcl-2, and B-cell lymphoma-extra large (Bcl-xL) proteins, along with the release of Cytochrome c (Cyto-C) and p53 upregulation, which in turn activates the caspase cascade and leads to cell apoptosis [6]. The extrinsic pathway involves the binding of ligands such as TNF-a Tumor necrosis factor-alpha (TNF-α), and the tumor necrosis factor-related apoptosis-inducing ligand (TRAIL) to their corresponding receptors followed by the formation of the death-inducing-signaling-complex [14]. The mechanisms of SR-induced apoptosis to treat tumors have been extensively investigated. Previous studies demonstrated that baicalein was capable of up-regulating the expression of Bax, reducing the expression of Bcl-2, and elevating the ratio of Bax/Bcl-2, decreasing mitochondrial transmembrane potential as well as triggering the release of Cyto-C into the cytoplasm, which in turn formed apoptotic vesicles with activated caspase-9 and apoptotic protease activating factor-1 (Apaf-1), thereby activating caspase-3 and the subsequent mitochondrial apoptotic pathway in human laryngeal carcinoma Hep-2 cells [15]. In support of this view, it was reported that baicalin treatment also up-regulated the mitochondrial Bax and cleaved caspase-3, while the expression of Bcl-2 protein levels was markedly decreased in a dose-dependent manner thus ultimately promoting apoptosis of pancreatic cancer SW1990 cell line. Flow cytometry cooperated with DAPI staining assay was also utilized to explore the level of apoptosis in SW1990 cells and indicated that baicalin significantly induced cell apoptosis dose-dependently. Research further demonstrated that baicalin promoted apoptosis in pancreatic cancer cells by activating Bcl-2 interacting mediator of cell death pathway. Consistently, the apoptosis-promoting effect of baicalin was also reported in HepG2 cells, and it is worth noting that the half-maximal inhibitory concentration (IC_50_) of baicalein was less than 40 μM [16]. A disproportional increase in mitochondria-mediated reactive oxygen species (ROS) was reported to be involved in cancer cell cycle arrest, senescence, and apoptosis. Increased mitochondrial oxidative stress leads to Cyto-C release, an irreversible event, and ultimately causes caspases activation and cell death [17]. Compared to normal cells, cancer cells exhibit extraordinarily high levels of ROS to alter redox status and contribute to the maintenance of malignant phenotypes [18]. Therefore, it is feasible to kill cancer cells selectively by promoting excess ROS production and inducing mitochondrial dysfunction [19]. Several studies revealed that baicalein treatment promoted ROS generation to induce mitochondrial dysfunction in multifarious cancer cells [20]. Based on this theoretical knowledge, Liu and colleagues provided evidence that baicalein, as a pro-oxidant, induced the mitochondrial pathway-dependent apoptosis and selective death of MCF-7 cells through mobilizing intracellular copper and forming a baicalein-copper complex, which was able to engage in redox cycling and lead to ROS overload [21]. These findings were further confirmed by a recent study, in which authors evaluated the cytotoxic and pro-apoptotic effects in myeloid leukemia K562 cells and found that all baicalein preparations induced the accumulation of ROS in K562 cells [22]. In addition, ROS also functions as the signaling molecule to activate the AMP-activated protein kinase (AMPK) pathway, thus inhibiting cell proliferation and tumorigenesis [23]. Wogonin has been reported to induce apoptosis by generating ROS and modulating p53 levels through AMPK activation, and oroxylin A, another major bioactive flavonoid in SR, also showed similar effects [24].

Today, there has been a growing awareness regarding the disruption of the endoplasmic reticulum (ER) homeostasis in the development and maintenance of cancer cells [25]. Disordered ER function further leads to IRE1α-TRAF2-ASK1 complex-meidated ER stress and mitochondria- and caspase-independent apoptosis. It was reported that increased expression of IRE1α in line with p-ASK was markedly reversed by wogonin treatmentwith (75 μM). Besides, Ge et al., reported that wogonin induced the expression and activation of ER chaperone genes like GRP78 and GRP94, as well as pro-death proteases which located on the outer surface of ER membrane, including caspase-4 and caspase-12, and thus exerted the pro-apoptotic function against cancer devolopment in malignant neuroblastoma cell lines (SK-N-BE2 and IMR-32 cells) [26]. Moreover, it is worth noting that the abnormal calcium signaling in ER is also involved in apoptotic responses [27]. Baicalein was repoted to induced Ca^2+^ accumulation by triggering phospholipase C-dependent Ca^2+^ release from the ER and Ca^2+^ entry via PKC-dependent, 2-amino-ethoxydiphenyl borate (2-APB)-sensitive Ca^2+^ channels, which ultimately induced ER-associated apoptosis in ZR-75-1 human breast cancer cells [28].

## 3. Regulation of Canonical Tumor-Associated Signaling Pathway

The major flavonoids in SR, baicalein, wogonin, along with their glycosides (baicalin and wogonoside) have been shown to possess several aspects of anti-cancer properties, which are involved multiple targets and pathways (Figure 1). Mutations in p53, a vital tumor suppressor, are reported in approximately 50% of all human malignancies [29]. A recent study suggested that wogonin increases the expression of p53 and p53-induced regulators of glycolysis and apoptosis and then by modulating p53-downstream glucose transporter 1 (GLUT1) and some key glycolytic enzymes such as Primary gastric melanoma (PGM), Hexokinase 2 (HK2) and phosphorylate pyruvate dehydrogenase kinase 1 (PDHK1), wogonin inhibits glycolysis in cancer cells and ultimately exert anti-cancer effect. The effect of wogonin on glycolysis was attenuated or even reversed in mutant p53 and p53-null cells both in vivo *or* in vitro, which further verified that the anti-glycolysis and anti-cancer effects were largely dependent on fully-functioning p53 [30]. Wang et al. explored the activity of ruthenium baicalein complex to efficiently target diverse apoptotic pathways in the murine model of colorectal cancer induced by 1,2-dimethylhydrazine and Dextran sulphate sodium (DSS). Cell-based reporter assay was conducted to analyze the effect of the ruthenium baicalein complex on the activity of p53, Vascular endothelial growth factor (VEGF), mammalian target of rapamycin (mTOR) protein kinase B (Akt), and caspase-3 signaling pathways. Both in vitro and in vivo studies provided robust confirmation that ruthenium baicalein complex possessed a potential chemotherapeutic activity to induce colon cancer apoptosis by activating p53-dependent intrinsic apoptosis and simultaneously down-regulating the AKT/mTOR and WNT/β-catenin pathways [31]. Cellular p53 levels promote the expression of gene of phosphate and tension homology deleted on chromosome ten (PTEN), a negative regulator of the PI3 kinase (PI3K) pathway that alters the cellular environment by decoding tyrosine kinase receptor signaling. Once PI3K is activated by PTEN, it assists the phosphorylation of Akt and activates mTOR pathways which together play a crucial part in tumor progression [32]. The PTEN/PI3K/AKT/mTOR pathway is suggested to be essential for the pro-apoptotic effects of SR flavonoids [33]. Multitudinous research has proved that baicalin was capable of inhibiting the phosphorylation in the PI3K/AKT pathway to facilitate the apoptosis of cancer cells [34]. Recently, a pair of chiral baicalin derivatives were synthesized by combining baicalin with phenylalanine methyl ester based on molecular docking technology, namely BAD and BAL. It was reported that their anti-tumor activities were relevant to the promotive effects on A549 cell apoptosis by inhibiting the phosphorylation of AKT, and BAL had higher antitumor activity than BAD [35]. Furthermore, the anti-tumor effects of baicalein against cervical cancer cells in line with xenograft tumor models in female athymic BALB/c nude mice were also attributed to the suppression of PI3K/AKT pathway. Yu and his colleagues identified a novel long noncoding RNA (BDLNR), which contributed to cervical cancer cell proliferation. RNA pull-down assays further demonstrated that BDLNR physically interacted with YBX1, a critical player in the transcription initiation of phosphatidylinositol-4,5-bisphosphate 3-kinase, catalytic subunit alpha (PIK3CA). Baicalein inhibits PIK3CA by downregulating the expression of BDLNR, and ultimately blocked the PI3K/AKT signaling and thus inhibited the growth of ovarian cancer in tumor-bearing mice [36].

Apart from PI3K/AKT signaling, increasing evidence suggests that the Wnt/β-catenin signaling cascade is a promising target of cancer treatment [37,38]. Xia et al. applied network pharmacology to investigate the potential targets and molecular pathways of baicalein in inhibiting the growth of colorectal cancer. They indicated that Wnt/β-catenin was a key pathway involved, and the feature protein β-catenin, glycogen synthase kinase 3β (GSK3β), and BAX were identified as the potential targets [39]. Baicalein also exhibited a potential inhibitory effect on the growth and development of cervical carcinoma via negatively regulating Wnt/β-catenin signaling, suppressing the transcriptional activity of β-catenin on pro-tumor genes like high-mobility-group box-2 (Sox-2), matrix metalloproteinases (MMP-2), and Nanog homeobox [40]. Moreover, investigations on the relationship between decidual protein induced by progesterone (DEPP) and tumor cell death have gradually become a hot spot [41]. The induction of DEPP increases the level of phosphorylated extracellular regulated protein kinases (ERK) and its target transcription factor Elk-1 [42]. Further experiments revealed that baicalein stimulated apoptosis and morphological changes of HCT116, Panc-1 and A549 cells via upregulating the mRNA and protein levels of DEPP and growth arrest and DNA damage-inducible 45α, which ultimately activated caspase-3, caspase-9 and the JNK/ERK/p38 MAPK pathways [43]. However, the reasons why baicalein induced DEPP expression and downstream MAPKs activation were still unclear, and the underlying mechanisms remain vague.

## 4. Inhibition of Cell Cycle Transition

Apart from the involvement of baicalein/baicalin in regulating the canonical tumor-associated signaling pathway, these flavonoid compounds exert their therapeutic effects against cancer by restraining cell cycle transition and survival at certain checkpoints (Figure 1). Cell proliferation is characterized by cytogenetic DNA replication and cell division, implemented by means of the cell cycle [44]. In actively dividing mammalian cells, the cell cycle is divided into four consecutive phases, including G1 (gap 1), S (DNA replication), G2 (gap 2), and M (mitosis-cell and nuclear division) phase [45], whereas the dysregulations lead to aberrant proliferation and eventually the malignant phenotype of cancer cells [46]. During G1 phase, cells decide either to enter the cell cycle; initiate DNA replication and divide; or to exit the cell cycle and enter quiescence, senescence, or differentiation stages. Genes involved in the G1/S transition have been classified into three categories: cyclin-dependent kinases (CDKs), CDK subunits (cyclins), and CDK inhibitors (CDKIs), as well as phosphorylated retinoblastoma protein. They coordinate with each other and form an elaborate regulatory network with cellular signal transduction pathways, thus together constituting the molecular basis of cell cycle regulation [47]. Extensive in vivo and ex vivo experiments have revealed that flavonoids from SR inhibited cell proliferation by blocking the cell cycle. A recent study found that baicalein severely impedes the progression of the G0/G1 transition in a dose-dependent manner, preventing the proliferation of HeLa and SiHa cells compared to the control group. Further studies suggested that baicalein arrested the cell cycle in the G0/G1 phase by downregulating the expression of the cyclin D1. As a target of AKT kinase, GSK3β has been reported to enhance the phosphorylation and degradation of cyclin D1, leading to further delay in cell cycle transition. Supporting this view, both p-AKT and p-GSK3b were downregulated in SiHa cells under the treatment of 40 µg/mL of baicalein. Furthermore, the inhibitor of GSK3β, CHIR-99021, reversed the downregulation of cyclin D1 induced by baicalein. To sum up, baicalein down-regulates the expression of AKT/GSK3β, thereby promoting the degradation of cyclin D1, and ultimately arrests the cells at G0/G1 phase [48,49]. Baicalin also blocked cell cycle at G0/G1 phase and inhibited the proliferation of acute B-lymphoblastic leukemia cells through the same mechanism [50]. S phase was also considered to be affected by baicalein, since in HepG2 and HepJ2 hepatoma cells, baicalein dramatically caused cell-cycle arrest early in S-phase by inducing DNA damage response, triggering the expression of CDK inhibitors such as p21 or p27 [46].

Moreover, G2/M phase was likely the primary target of flavonoids from SR. Kuo et al., demonstrated that baicalein induced morphological changes and decreased the number of HCC J5 cells at M phase through the induction of G2/M arrest, and also triggered J5 cells apoptosis in a dose- and time-dependent manner at the same time. Protein levels associated with G2/M phase, including cell division cycle 25C (Cdc25c), cell division cycle 2 (Cdc2) in line with cyclin B1, were all decreased [51]. Interestingly, another study examined the phosphorylation level of CHK1, a marker of DNA damage response for S-to-G2/M phase arrest and demonstrated that baicalein inhibited the proliferation of head and neck squamous cell carcinoma HSC-3 cells, and arrested cells at G2/M phase [52]. In addition, wogonin was reported to be involved in regulating the cell cycles of colorectal cancer cells, significantly increasing the percentage of SW48 cells in G2 phase from 10.12 to 48.15% [53]. In bladder cancer cells, baicalein treatment induced G2/M arrest by decreasing the expression of cyclin B1 and phospho-Cdc2 (Thr161), two vital proteins related to mitosis initiation [54]. In addition, cyclin A was also reported to act as a core regulator of DNA replication and mitosis in the G2/M phase. Cyclin A binds to CDK2 (cyclin-dependent kinases 2), which results in the inactivation of its DNA helicase activity and prevents DNA re-replication, thereby arresting the cell cycle in G2/M phase [55]. Baicalein along with other extracts halted the cell cycle during the G2/M-phases in the human colorectal cancer cell lines, such as HCT-116 and HT-29 cells, by promoting the expression of cyclin A [56].

## 5. Inhibition of Tumor DNA Damage Repair

DNA damage and repair (DDR) is the fundamental hall-mark of cancer that increases genomic instability and mutagenesis, acting a dominating role in malignant progression [57]. Mutagenesis tends to cause the maladjustment of gain-of-function mutations in oncogenes, or loss-of-function mutations in tumor-suppressor genes, which ultimately evokes neoplastic growth [58]. Telomerase is a ribonucleoprotein complex containing hTP1, hTR, and hTERT, which acts in the maintenance of telomere to promote neoplastic growth and extend lifespan of cancer cells [59]. Huang et al., found that wogonin downregulated the transcription of both hTERT and hTP1, resulting in progressive telomere shortening and HL-60 leukemia cells death. Furthermore, wogonin also downregulated the hTERT promoter, Myc proto-oncogene protein, to suppress cancer cell proliferation [60]. Due to the similarity in the molecular structures between wogonin and baicalin, it is therefore conceivable that baicalin may exert potential effect on the growth of HL-60 cells via the same mechanism as wogonin. A recently report also found that baicalin inhibited the proliferation of HL-60 cells by down-regulating c-Myc along with its target gene [61]. Genomic instability is an underlying hallmark of cancer, which is associated with a greater propensity to DDR [62]. DNA damage triggers a sequence of cellular signaling pathways and ultimately results in the DNA breaks and apoptosis [63]. The DDR mechanisms are involved in divergent types of damage, containing base excision repair, nucleotide excision repair, mismatch repair, homologous recombination repair, in line with non-homologous end joining [64]. Poly (ADP-ribose) polymerase (PARP) is a key substrate of activated caspase-3, and the use of its inhibitors has shown profound benefit against homologous recombination repair [65]. Chao et al. reported that baicalein treatment dose-dependently induced Ewing’s sarcoma cell apoptosis by causing a remarkable increase in the cleavage of PARP [54]. Supporting in this view, baicalein combined with chloroquine was also found to apparently increase PARP cleavage and reduce cancer cell viability in both HEY and A2780 ovarian cancer cell lines [66]. Furthermore, there was evidence supporting that wogonin exhibited anti-hepatocellular carcinoma effects through the inhibition of PARP [67].

## 6. Inhibition of Tumor Metastasis and Angiogenesis

Metastasis accounts for approximately 90% of cancer-related complications and treatment failure [68]. Thus far, metastasis involves a series of interrelated processes, including invasion, survival, arrest, as well as metastatic colonization. Several mechanisms are involved in cancer invasion and metastasis, such as epithelial-mesenchymal transition (EMT), extracellular matrices (ECM) degradation, and angiogenesis.

EMT is a cellular process in which cells lose their epithelial characteristics and acquire mesenchymal features, and is strongly associated with various tumor functions, including tumor initiation, malignant progression, tumor stemness, tumor cell migration, intravasation to the blood, metastasis, and resistance to therapy [69]. Epithelial markers such as E-cadherin and mesenchymal markers including vimentin and fibronectin. Baicalein was found to alleviate EMT and inhibit the migration and invasion of breast cancer cells in a time- and dose-dependent manner by inhibiting both Wnt/β-catenin pathway as well as SATB homeobox 1 [70]. Coincidentally, Zhou and his colleagues utilized a xenograft metastasis tumor model of breast cancer cells to investigate the anti-tumor effect of baicalin in vivo. They demonstrated that baicalin treatment decreased β-catenin expression and reversed the EMT process, with an exchange from a mesenchymal feature to an epithelial phenotype in breast cancer cell lines. Furthermore, an adenovirus vector system overexpressing β-catenin significantly abolished the inhibitory effects of baicalin on metastasis and EMT in breast cancer cells, supporting the former view [71]. Similarly, wogonoside was also reported to attenuate EMT in cutaneous squamous cell carcinoma and the potential mechanism was associated with the inhibition of β-catenin signaling pathway in a recent study. The upregulated expression of E-cadherin and the downregulated expression of vimentin and fibronectin were observed under the treatment of 100μM wogonoside and were partially antagonized by skl2001, a β-catenin agonist [72].

To date, in addition to Wnt/β-catenin pathway mentioned above, numerous studies have demonstrated that multiple signaling pathways also participate in the regulation of EMT, like transforming growth factor beta 1 (TGF-β1)/p-SMAD Family Member 3 (Smad3) and p38MAPK signals [73]. TGF-β1, one of the members of the transforming growth factor family, is highly expressed in tumor cells [74]. In order to investigate the potential role of baicalin in inhibiting the migration and invasion of colorectal cancer cells, transwell assay, wound-healing assay as well as immunoblotting were carried out. The results showed a decrease of TGF-β1 pathway-related proteins followed by a significant suppression of mesenchymal- associated markers associated with the therapeutic effects of baicalin [75]. Similarly, baicalein also markedly inhibited EMT induced by TGF-β1 and thus inhibited the migration and invasion of human A549 lung adenocarcinoma cells and osteosarcoma cells [76]. The above evidence suggested that flavonoids in SR inhibited EMT to suppress tumor metastasis through multiple pathways.

Tumor angiogenesis promotes malignancy progression via constant supplement of oxygen and nutrients and functions in response to signaling such as the VEGF [77,78]. It has been reported that baicalein exerted anti-angiogenic effects by up-regulating p53, inhibiting mTOR activation and suppressing VEGF expression which further induced angiogenesis and survival of cancer cells [68]. Emerging evidence also suggested that baicalin reduced the expression and activity of multiple cytokines, including VEGF and MMPs, suppressed angiogenesis, and had promising impacts on cancer cell viability and proliferation both in vitro and in vivo. Wogonoside was reported to suppress VEGF production in MCF-7 cells. Wogonoside inhibits VEGF production in MCF-7 cells and blocks breast cancer angiogenesis by downregulating the wnt/β-catenin pathway. By assessing the effects of β-catenin nuclear accumulation, they suggested that wogonoside prevented β-catenin translocation to the nucleus and downregulated its binding to T-cell factor/lymphoid enhancer-binding factor (TCF/Lef), the major effector of the Wnt/β-catenin signaling pathway, which ultimately inhibited the activation of target genes such as VEGF to halt angiogenesis in breast cancer [79]. A possible mechanism of inhibition of anti–tumor metastasis and angiogenesis is indicated (Figure 2). In addition to the suppression of breast cancer, studies also found that wogonoside inhibited the invasion and migration of lung cancer A549 cells by targeting angiogenesis of xenograft tumors in nude mice, providing a theoretical basis for the usage of wogonoside in the clinical treatment of lung cancer [80].

## 7. Regulation of the Immune-Related Tumor Microenvironment

Emerging evidence suggests that except for neoplastic cells arranged in the compact nests, carcinoma tissues also present a significantly altered surrounding stroma [81]. Initially regarded as a disease related to disordered genetic and cellular expression, cancer is further regarded as the result of the TME disorder [82]. TME is a complex and rich multicellular environment for the survival and development of cancer cells, consisting of various types of cells mediating the communication between tumor cells, inflammatory immune cells (such as macrophages and their precursor monocytes), other non-malignant stromal cells including fibroblasts and mesenchymal cells and extracellular matrix (ECM) [83,84]. Emerging evidence suggested that SR regulated TME by three major categories: inflammation, hypoxia and immunosuppression [85]. The major flavonoids in SR governed regulation of the immune-related tumor microenvironment properties via different mechanisms as illustrated in Figure 3.

Inflammation plays not only a causative role but also serves as the consequence of cancer, since the inflammatory environment accelerates the formation of the tumor microenvironment and facilitates genomic mutations and other oncogenic effects. A vast body of epidemiological studies has implicated that inflammation enhances the initiation, progression, and aggressiveness of various malignancies. As the major regulator of inflammatory responses, Nuclear factor-k-gene binding (NF-κB) was reported to drive the generation of inflammatory events associated with cancer progression [86]. Recent studies demonstrated that wogonin exerted immunoregulatory effects via attenuating TNF-α conferred NF-κB activity and thereby sensitized malignant T cells to induce apoptosis [81]. NF-κB Interacting LncRNA (NKILA) is a recently identified lncRNA, which physically interacts with NF-κB/IκB, inhibits IκBα phosphorylation and NF-κB nuclear translocation, and therefore decreases NF-κB activity in cancer cells. When the overexpressed NKILA vector was utilized in combination with 50 μM baicalin, the viability of HCC cells decreased about 3-fold compared with the same dose of baicalin alone. The results suggested that overexpression of NKILA could enhance the inhibitory effect of baicalin on HCC. Furthermore, knockdown of NKILA significantly reversed the inhibitory roles of baicalein on NF-κB [87]. Besides, the relationship between NF-κB and oxidative stress in tumor cells has also been reported. Baicalein treatment significantly inhibited leukemia via attenuating vascular endothelial cell loss, edema, inflammatory cell infiltration, and blood clots, as well as reducing the serum levels of TNF-α, interleukin 1β (IL-1β), interleukin 6 (IL-6), and intercellular cell adhesion molecule-1 (ICAM-1). Further tests in vitro demonstrated that baicalein lessened endothelial cell apoptosis, decreased intracellular ROS levels, suppressed phosphorylation of p38, and eventually inhibited activation of NF-κB signaling pathway [88].

The immune system plays dual roles in tumor development. On one hand, the body controls tumor occurrence and development by driving innate and acquired immunity. On the other hand, proteins produced by the tumor cells often render an immunosuppressive microenvironment to facilitate tumor immune escape [89,90,91]. Among diverse inflammatory cells infiltrating the TME, tumor-associated macrophages (TAMs) predominantly play a leading role in the formation of the immunosuppressive TME [92]. Although M1 macrophages are usually considered anti-tumor, the majority of TAMs are possessing M2-like phenotype and are tumor-promoting [93]. Therefore, controlling the repolarization of M2-like TAMs to M1 phenotype is conducive to advance the development of innovative cancer immunotherapy [94]. Accumulating evidence supported that bioactive flavonoids in SR exerted antitumor effects by modulating the TAM-associated TME. A recent study suggested that a combination of flavonoids in SR (including wogonin, baicalein and baicalin), termed WBB, regulated macrophage polarization and antitumor immune responses in non-small cell lung cancer (NSCLC). First, the systems pharmacology approach was utilized to predict active components of SR targeting macrophages in TME through compound-target and target-microenvironment phenotypic association analysis. In line with the results of bioinformatics analysis, in vivo and in vitro experiments demonstrated that WBB triggered M1 polarization in macrophages and induced an M2 towards to M1 phenotype transition in a concentration-dependent manner. Upon further exploration of the molecular mechanisms mediating this effect, the researchers demonstrated that WBB remarkably increased the expression level of Janus kinase2 (JAK2) and STAT1 Signal Transducers and Activators of Transcription 1 (STAT1), suggesting that WBB could target the JAK2-STAT1 pathway. It is university acknowledged that the JAK2-STAT1 pathway act as a core player in modulating IFN-related immune responses, and STAT1 is emerged as a key mediator which regulates the activation of macrophage M1/M2 phenotypes. The addition of a STAT1 inhibitor resulted in a significant reduction in the percentage of M1 macrophages in the cells (from 46 to 23%), which confirmed that WBB regulated M1 polarization of macrophages via the STAT1- dependent signaling pathway [95]. For individual flavonoid in SR, by using an orthotopic hepatocellular carcinoma implantation mouse model, Tan et al. found that, mechanistically, baicalin initiated autophagic degradation of TRAF2 in TAMs and activated the p52 pathway, which thereby induced the polarization of M1-like macrophages and halted HCC development by remodeling the immunosuppressive TME [96]. A recent study encapsulated baicalin with Mannose (Man) to bind to Macrophage mannose receptor receptors on the surface of TAM, aimed to investigate whether baicalin could reprogram macrophages toward an M1-like phenotype. Results showed that the modified baicalin exerted tumor suppression effects, with a 6-fold upregulation of IL-6 and TNF-α in the supernatant of RAW264.7 cells compared with the control group. By presenting a suitable surface charge, the nano-compound was more stable in systemic circulation and was then preferentially taken up and aggregated at tumor sites [97]. Likewise, Han et al., successfully assembled a bioactive nano-complex in which PLGA nanoparticles were fabricated to encapsulate baicalin. When further modified with specific ligand M2pep and α-peptide ligands, the nano-complex selectively targeted m2-like TAMs and successfully reversed their phenotypes to M1-like phenotypes. As a result of improved TME, substantial retardation of tumor growth was observed in tumor-bearing mice after intravenous administration of the nano-complexes for one week. In in vitro experiments, the nano-complex activated both M1-like and M2-like macrophages, further resulting in greater secretion of inflammatory cytokines such as IL-12, IL-10, IL-2, and IFN-γ as well as the activation of T cells [4]. in addition to baicalin and baicalin-based nanoparticles, it has been reported that baicalein not only directly targeted breast cancer cells but also inhibited MDA-MB-231 and MCF-7 breast cancer cell growth via promoting repolarization of M2-TAM to M1 phenotype and attenuating TGF-β1 secretion [73].

Furthermore, baicalein and baicalin have been reported to inhibit the activity of STAT3. As a convergence point for numerous oncogenic signaling pathways, STAT3 is not only broadly involved in numerous biological processes, but also acts dominant role in suppressing the expression of crucial immune activation regulators. Therefore, it is not surprising that STAT3 has emerged as a promising tumor therapeutic target [98,99]. Yang et al., reported that wogonin activated NF-κB and suppressed STAT3, and thus suppressed tumor cell growth by promoting macrophage M1 polarization in vitro, and increasing immune cell infiltration in xenografted tumors in vivo [100]. In another study, in addition to direct cytotoxicity, baicalein and baicalin stimulated T cell-mediated immune response against tumors by reducing programmed death-ligand 1 (PD-L1) expression in SMMC-7721 and HepG2 cells. Further mechanistic research indicated that baicalein and baicalin decreased STAT3 activity, down-regulated IFN-γ-induced PD-L1 expression, and subsequently restored T cell sensitivity to eliminate tumor cells [101].

## 8. Applications of Flavonoids in SR in Reversing Drug-Resistance and Sensitizing Chemotherapy

Naturally derived compounds are increasingly tested as adjuvant therapy in cancer treatment due to their synergistic effects with chemotherapeutic drugs or targeted therapies, which tend to enhance their efficacy at low doses as well as to minimize potential adverse effects to improve patient compliance and quality of life [102]. Multi-drug resistance to conventional therapies is one of the main causes of chemotherapy failure and is still regarded as the major obstacle to the clinical treatment of cancers. Increasing evidence suggested that baicalein potentiated the efficacy of chemotherapy in anticancer therapy. Hyperactive aerobic glycolysis is the hallmark of cancers and contributes to the development of resistance to therapeutic agents. Recent findings by Chen and his colleagues suggested that baicalein significantly enhanced the oral-bioavailability of tamoxifen (TAM). The nuclear translocation and transcriptional activity of hypoxia-inducible factor-1α (HIF-1α) were reduced by baicalein, which contributed to the downregulation of HIF-1α–mediated glycolysis uptake and recovered the number and shape of mitochondria as well as their ability to generate ROS, and thus re-sensitized breast cancer cells to TAM treatment [103]. As a critical chemotherapeutic agent currently utilized in the treatment of a wide range of cancers, cisplatin exerts its cytotoxic effects in tumor cells primarily through the production of deoxyribonucleic acid-platinum complexes and the subsequent deoxyribonucleic acid damage response [104]. Xu et al., evaluated the effects of combining baicalin and cisplatin (DPP) on the proliferation of A549 and A549/DPP (DPP-resistant) human lung cancer cells by MTT assay, and suggested that baicalin exerted synergistic anti-tumor effects by reversing the resistance of lung cancer to cisplatin, and the underlying mechanism was related to the down-regulation of both Microtubule affinity-regulating kinase 2 (MARK2) and p-AKT [34]. Likewise, baicalein was also reported to significantly reduce the expression of MARK2 and p-Akt in cisplatin-resistant cells in a dose-dependent manner, by decreasing the expression of drug-resistant proteins such as human multidrug resistance 1 (MDR1), multidrug resistance protein 1 (MRP1), and excision repair cross-complementation 1 (ERCC1), and thus suppress both drug-efflux and cell proliferation [105].

Coincidentally, the reversal of multidrug resistance by wogonin has attracted the attention of researchers recently. It has been reported that the combination of wogonin and cisplatin circumvented cisplatin resistance and restored the cytotoxic effect of cisplatin in head and neck cancer cells. Further investigation revealed that this effect was conducted by inhibiting nuclear factor erythroid 2-related factor 2 (Nrf2)-mediated cellular defense responses in coordination with depleting glutathione and transferases such as glutathione *S*-transferase pi. Besides, wogonin also reduced the export of anticancer agents from cancer cells via inhibiting P-glycoprotein drug efflux pump [106]. Additionally, wogonin also reverses drug resistance by modulating various pathways. Hong et al. investigated whether wogonin could enable a more moderate use of cisplatin. In particular, the cytotoxicity of 10 μmol/L cisplatin with the presence of 50 μmol/L wogonin was comparable to that of high-concentration cisplatin (100 μmol/L) in BGC-823 cells. Further investigations of the mechanism demonstrated that wogonin in combination with cisplatin increased the phosphorylation of JNK, thus enhancing cisplatin-induced apoptosis and autophagy in human gastric cancer cells [107]. Recently, wogonin was also reported to inhibit PI3K/AKT signaling pathway to reverse cisplatin resistance in ovarian cancer SKOV3 cells [108]. Inspired by these findings, Qin designed hybrid compounds by conjugating a wogonin unit with platinum (IV) complexes to form a new compound [109]. As a multi-targeted anticancer agent, the new complexes not only showed improved antitumor properties by acting on multiple targets compared with cisplatin, but also exerted strong anti-inflammatory and antioxidant activity from the advantage of wogonin. Thorough mechanistic research indicated that integrative complexes exerted anti-cancer effects by activating the p53 signaling pathway, leading to the accumulation of intracellular ROS, and inducing mitochondrial damages and apoptosis.

## 9. Discussion

Although extensive scientific literature reported the promising anti-malignant potential of flavonoids in SR, such as baicalein and wogonin, the poor water solubility and lipid solubility of these ingredients lead to poor absorption in the gastrointestinal tract and restrict their clinical application. A recent in vivo study found that after oral administration of baicalin and baicalein (both at 224 μM/kg) in rats, the absolute bioavailability was 2.2% and 27.8%, respectively [110]. Pharmacokinetic studies showed that baicalin is unstable in the intestine and is metabolized by β-glucuronidase or intestinal microflora to release its glycoside baicalein [111]. Baicalein is then converted back to baicalin or baicalein 7-O-sulfate, catalyzed by UDP-glucuronosyltransferase (UGT) in the small intestine and liver [112]. Similarly, wogonin is metabolized to wogonin-7-D-glucuronide in the liver via glucuronide incorporation, resulting in the observation of a mass of wogonin-7-D-glucuronide and a tiny proportion of wogonin in the bloodstream, which is related to the low absorption of wogonin [113].

It is especially important to improve the bioavailability of these bioactive flavonoids by using different techniques to broaden the prospect of their clinical application. With the advent of nanotechnology, baicalein-coupled nanoparticles have shown promising results in in vivo and in vitro studies. Recent reports demonstrated a formulation (BCN-SNS), prepared under high-pressure homogenization of a mixture of baicalein, microcrystalline cellulose, and carboxymethylcellulose sodium (MCCS), and d-α-tocopherol polyethylene glycol 1000 succinate (TPGS), significantly enhanced the oral bioavailability and the dissolution rate of baicalin. In BCN-SNS-treated rats, the maximum blood concentration (C_max_), AUC, and elimination half-life (t_1/2_) were increased approximately 3-fold, 7-fold, and 3-fold, respectively compared with coarse baicalein. Meanwhile, the clearance was reduced by approximately 7-fold [114]. Not coincidentally, Harsh and his colleagues fabricated and characterized in situ solid lipid nanoparticles of baicalein (SLNB) by the coacervation method, where baicalein was incorporated into a crystalline state, leading to more efficient drug entrapment and release kinetics. Pharmacokinetic studies revealed that along with a 313% increase in oral relative bioavailability, SLNB also exhibited higher AUC, T_1/2_, and mean residence time than the free baicalein. Further, after 4h incubation with murine splenic lymphocytes followed by exposure to 4 Gy dose of radiation, 25 μM SLNB is approximately 3-fold more efficient in protection than equimolar free baicalein. At the meantime, SLNB (25 μM) seemly sensitized A549 lung cancer cells to radiation-induced death by 5 folds. This enhanced in vitro radioprotective efficacy of SLNB over free baicalein may be attributed to increased stability and cellular uptake of baicalein in the nano-formulation form [115]. However, the mechanism underlying the selectively radioprotective effects of SNLB on normal cells is yet to be elucidated by further investigation. It was also reported that baicalein could reverse multidrug resistance in cancer treatment by association with nanocrystals. Researchers prepared a novel nanoemulsion (PTX/BA NE) for co-delivery of paclitaxel (PTX) and baicalein to enhance anti-tumor effect. PTX and baicalein were encapsulated in a complex with cholesterol or phospholipid to improve the solubility and stability in the oil phase of nanoemulsions. The combinational effects of PTX and baicalein with diverse weight ratio were evaluated on Taxol-resistant of human breast cancer cell line MCF-7/Tax cells by MTT assay, and the highest synergistic effect was exhibited when PTX and baicalein were combined at a mass ratio of 1/1. Importantly, this complex not only exerted intensive cytotoxicity in vitro but also facilitated cancer-specific distribution when compared with either free PTX or PTX NE in MCF-7/Tax cells. The in vivo evaluation showed that the tumor inhibition rate of PTX/BA NE was 77.0%, which was significantly more effective than other treatments. Mechanism research demonstrated that, in response to PTX/BA NE treatment, the intracellular ROS was increased, GSH was depleted, and caspase-3 dependent apoptosis was activated [116]. Since the wogonoside shows poor plasma concentration and bioavailability after oral administration [117], various excellent attempts were made to improve the bioavailability of wogonin. Suep Baek et al., optimized the wogonin-loaded solid lipid nanoparticle (W-SLN) with stearic acid, lecithin, poloxamer and mannitol at a mass ratio of 1/10/7.5/7.5/2.5. Compared to the wogonin solution and blank solid lipid nanoparticle, W-SLN exhibited the highest cellular uptake (12.8 ± 2.1 ng/µg) by MCF-7cells, increasing about 3 folds. A drug release study also proved that wogonin release was at around 84.6 ± 4.0% from the W-SLN even after 72 h, while wogonin solution complete release at only 4 h [113]. In another study, Tian and his colleagues formulated wogonin liposomes modified with glycyrrhetinic acid to target liver cancer actively and specifically. Compared with free wogonin in solution, the compound exhibited significant improvement in the biodistribution, accumulation in line with therapeutic efficacy [118].

The use of Chinese herbal medicines (CHM) in combination was reported to be another promising approach to effectively improve the pharmacodynamic and pharmacokinetic properties of flavonoids in SR. Zhang et al., suggested that CHM with clearing heat and promoting diuresis properties significantly improved the absorption of baicalin without affecting its elimination process [119]. Based on SR−*Coptidis rhizome* (CR, *Huanglian* in Chinese) combination, inspired by the drug combination in TCM, Lei and his colleagues discovered that berberine (BBR), a dominant component of CR, integrated with baicalin (BA) to formulate natural self-assemblies as BA-BBR nanoparticles (BA-BBR NPs), showing synergistic effects against Diarrhea-predominant irritable bowel syndrome IBS-D. In vivo experiments were performed, and compared with the free BA, BA-BBR NPs showed a better therapeutic effect on IBS-D, suggesting that the self-assembled BBR significantly improves the therapeutic effect of baicalin [120]. Although the investigation did not continue to test other pharmacological effects of these BA-BBR NPs, it is undeniable that finding supramolecular nano-formulations according to the theory of TCM compatibility will provide novel approaches to improve the pharmacokinetics and pharmacodynamics properties of flavonoids in SR.

According to clinical symptoms, TCM theory attributes the pathogenesis of cancers to fire and heat attack and dampness, which has led to the use of heat-clearing, detoxification, and heat-dampness-clearing formulas for the treatment [121]. SR has been considered as a representative medicine for clearing away heat and fire in several ancient documents. It has been well accepted that the scientific connotations of clearing heat and fire are coinciding with the principle of modern medicine for the regulation of immune responses and probably also gut microbiota. Unquestionably, the mechanism of TCM by regulating gut microbiome-tumor immunity axis has become a trend in the field of future scientific research and might hold clues to more informed decisions to unlock a new era of cancer immunotherapy [122]. With the deepening comprehension of gut dysbiosis, emerging data suggested that gut microbiota modulate the effectiveness of cancer therapies, especially immunotherapy [123,124]. On the one hand, dysbiosis of intestinal flora exerts pro-inflammatory effects by directly driving microorganism-associated molecular patterns-mediated Toll-like receptors signaling and thereby facilitates carcinogenesis [125]. On the other hand, the gut microbiota also helps to boost the efficacy of cancer immunotherapies. Extensive research has shown that intestinal microbes stimulate the expression of chemokines such as TNF- α, IL-6, IL-10, and IL-1β or activate cytotoxic lymphocytes, such as CD4^+^ and CD8^+^ T cells or NK and NKT cells, in both peripheral blood and tumor to limit tumor growth [126]. Furthermore, emerging evidence also reported that some “beneficial bacteria” could also be utilized as “immune enhancers” to assist immunotherapy [127]. Baicalin has been reported to regulate intestinal flora and effectively alleviate intestinal barrier damage by promoting the production of SCFAs [128]. Consistently, Zhu et al., also reported that baicalin suppressed colorectal adenoma growth at doses ranging from 25 to 100 mg/kg by regulating intestinal flora (butyrate-producing bacteria such as *Butyricimonas* spp., *Roseburia* spp., *Subdoligranulum* spp., and *Eubacterium* spp.), which provides new insights into the pharmacological effects and therapeutic mechanisms of baicalin [129]. The network linking flavonoids in SR, gut microbiota, and tumor immunity needs to be elucidated.

## 10. Conclusions

SR, a broadly used herb in China as a heat and fire clearing agent, has been demonstrated to exhibit potential anti-tumor effects against various types of cancers in both in vitro and in vivo experiments. Emerging evidence has suggested that flavonoids are prominent bioactive components in SR possessing anti-cancer activities through the induction of cancer cell apoptosis, inhibition of cancer cell proliferation, metastasis as well as angiogenesis, regulation of the tumor microenvironment, and reversing drug resistance. Although the pharmacological studies at present exist deficiencies as described above, it is still reasonable to conclude that SR and its bioactive flavonoids have a promising prospect as adjuvant or even primary therapies for the clinical management of cancers.

## Figures and Tables

**Figure 1 ijms-23-11042-f001:**
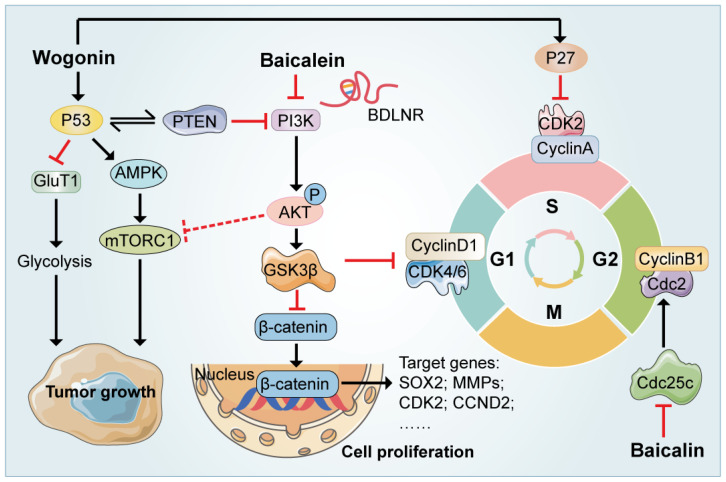
Mechanism of baicalein, wogonin, along with their glycosides (baicalin and wogonoside) in inducing cell cycle arrest and regulating the canonical tumor-associated signaling pathway. Baicalin and wogonin induced cancer cell cycle arrest by regulating cyclin proteins (such as cyclin A, cyclin B1 and cyclin D1). Meanwhile, baicalein and wogonin promote cancer cells apoptosis and inhibit proliferation via PI3K-AKT and AMPK-mTOR pathways, respectively.

**Figure 2 ijms-23-11042-f002:**
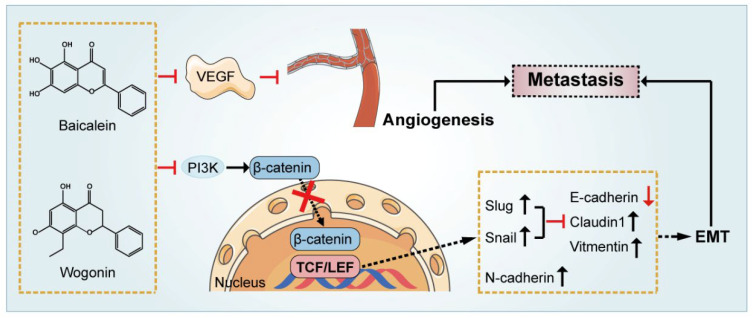
Cellular targets of baicalein and wogonin in modulating tumor cells angiogenic and metastasis. On one hand, baicalein and wogonin directly decreasing the level of vascular endothelial growth factor, thereby suppressing cancer cell angiogenesis. On the other hand, by inhibiting the nuclear translocation of β-catenin and T-cell family binding affinity, baicalein and wogonin eventually reverse the epithelial-mesenchymal transition effect. The black arrows represent up-regulated expression of proteins, and the red arrows represent down-regulated ones. Red cross indicates inhibition.

**Figure 3 ijms-23-11042-f003:**
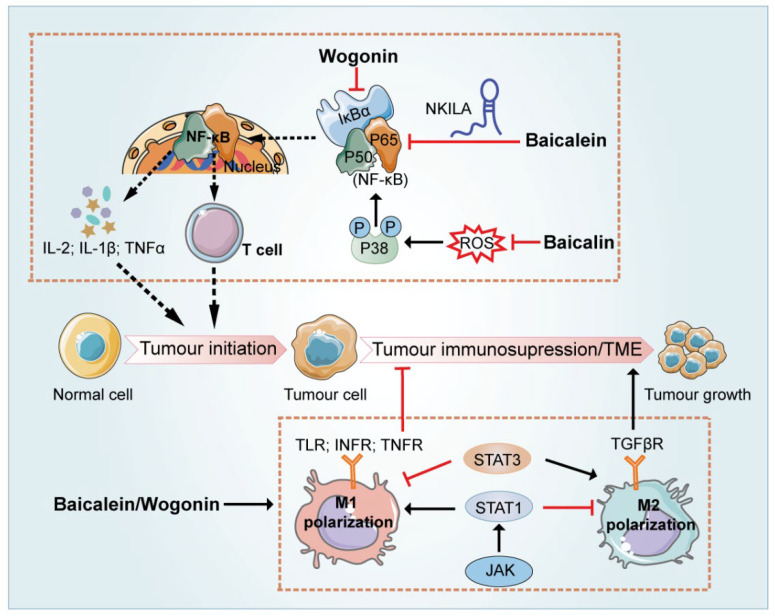
Molecular signaling networks underlying the anti-inflammatory and immunoregulative effects of baicalein, wogonin, along with their glycosides (baicalin and wogonoside). Baicalein promotes the physical interaction between NF-κB interacting LncRNA and NF-κB/IκB, inhibits IκBα phosphorylation and NF-κB nuclear translocation, therefore decreases NF-κB activity and exerts anti-inflammatory effect against tumors. Baicalin and wogonin improve immune and inflammation abnormality through the blockade of pro-inflammatory signaling pathways (such as TLR/NF-κB and JAK/STAT3), induction of cancer cell apoptosis, as well as macrophage reprogramming from an M2 to an M1 phenotype.

## Data Availability

Not applicable.

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
