# Peer review of "Advances in Anti-Cancer Activities of Flavonoids in *Scutellariae radix*: Perspectives on Mechanism"

_ijms, 2022, doi:10.3390/ijms231911042_

Round 1

Reviewer 1 Report

Good review exploring the application in cancer therapy of  Scutellariae Radix, from molecular perspective to clinical approach.

I suggest to check the english to avoid some typos, as well as to reduce Discussion section.

Reviewer 2 Report

My manuscript submitted for review describes anti-cancer activities of flavonoids in Scutellariae Radix. It is a detailed review of the scientific literature, mostly from the last 5 years, dealing with the anti-cancer activities of baicalin, baicalein, wogonin and wogonoside. The manuscript is very well structured, sequentially describing the role of these flavonoids in induction of cancer cell apoptosis, regulation of canonical tumor-associated signaling pathway and the immune-related tumor microenvironment, as well as inhibition of cell cycle transition and tumor metastasis and angiogenesis.

In this way, the authors substantiate the possibility that the considered flavonoids may find application in cancer therapy together with known chemotherapeutics or even alone.

I suggest that authors refine the title to prepare the reader for what they are about to read.

And a very minor remark - The letter "a" is missing in the word "Baicalin" in figures 1 and 3.

Reviewer 3 Report

Dear Authors,

the paper is very nicely written and covers a broad area of the anti-cancer activities of flavonoids of Scutellariae radix species. The Figures, which the authors have included in the paper, complement the text very well and make it easier to understand. The only thing I might suggest the authors do is write a little more about the plant itself and flavonoids in general, as well as baicalein and wogonin.
I would suggest the authors read the paper a bit and correct some typos. Also, I would note that they use a consistent font throughout the paper (section 2. starting with: "Increasing evidence...")

Reviewer 4 Report

Manuscript ID: ijms-1827883

The manuscript of Yiqing Gu et al. is interesting and very well written and describes the anticancer properties of several flavonoids, including baicalein, wogonin, baicalin and wogonoside. The authors clearly describe the targets and molecular pathways regulated by the phytochemicals of Scutellariae Radix, focusing the Manuscript on the Hallmarks of Cancer modulated by these bioactive compounds, like evasion of the apoptotic pathways, evading growth inhibitors, pro-inflammatory tumor microenvironment, increased angiogenesis, activated invasion and metastasis, evading immune destruction and mechanisms of multi drug resistance. The authors should answer to the following Major and Minor Revisions in order to improve their Manuscript.

MAJOR REVISIONS

1)The authors should add an additional chapter for discussing the antitumor properties of baicalein, wogonin, wogonoside and baicalin regarding the other hallmarks of cancer, in particular the limitless replicative potential exerted through the overexpression and increased activity of hTERT and also the genome mutations which lead to cancer development and are caused by the inhibition of  the DNA damage repair mechanisms (NHEJ, BER, NER).

2)In the “Discussion section” the authors should discuss the use of nanotechnologies to improve the bioavailability of also the flavonoids wogonin and wogonoside.

MINOR REVISIONS

1)In the Graphical Abstract the authors should replace Bal-2 with Bcl-2 and Baiclin with Baicalin.

2)Are the phytochemicals baicalein, wogonin, baicalin, wogonoside able to activate the endoplasmic reticulum-associated pathway of apoptosis in cancer cells?

3)The molecular pathway modulated by Baicalein in Figure 1 is not clear: it seems that Baicalein, inhibiting PI3K, induces the nuclear translocation of β-catenin, which is an oncoprotein.

4)In the Discussion section , I think that the authors should remove the statements from “According to clinical symptoms,…..” to “… and probably also gut microbiota”, because these statements do not add any useful and innovative informations to the Manuscript.

Round 2

Reviewer 4 Report

The authors have answered to all my major and minor requests and remarkably improved their manuscript which, in my opinion, can now be accepted for publication in  International Journal of Molecular Sciences.